# Cytokine Pathways in Cardiac Dysfunction following Burn Injury and Changes in Genome Expression

**DOI:** 10.3390/jpm12111876

**Published:** 2022-11-09

**Authors:** Jana E. DeJesus, Jake J. Wen, Ravi Radhakrishnan

**Affiliations:** Department of Surgery, University of Texas Medical Branch, 301 University Blvd., Galveston, TX 77550, USA

**Keywords:** burn injury, cytokines, cardiomyopathy, heart dysfunction, transcriptome

## Abstract

In 2016, an estimated 486,000 individuals sustained burn injuries requiring medical attention. Severe burn injuries lead to a persistent, hyperinflammatory response that may last up to 2 years. The persistent release of inflammatory mediators contributes to end-organ dysfunction and changes in genome expression. Burn-induced cardiac dysfunction may lead to heart failure and changes in cardiac remodeling. Cytokines promote the inflammatory cascade and promulgate mechanisms resulting in cardiac dysfunction. Here, we review the mechanisms by which TNFα, IL-1 beta, IL-6, and IL-10 cause cardiac dysfunction in post-burn injuries. We additionally review changes in the cytokine transcriptome caused by inflammation and burn injuries.

## 1. Introduction

Burn injuries trigger the systemic release of inflammatory mediators, contributing to organ dysfunction at distant sites from the original injury. In the heart, left ventricular function has been found to decrease within 24 h of burn injury and remain depressed for up to 72 h [1]. Cytokines play a role in promoting this inflammatory process and have been found to be higher in concentration in cardiomyocytes compared to the systemic circulation immediately post burn injury [2]. While there are numerous studies on the molecular mechanisms of cytokines in burn-induced cardiac injuries, there is a limited knowledge on changes in the cytokine transcriptome in cardiac dysfunction following burn injuries.

Cardiac dysfunction has been associated with negative patient outcomes in burns. In pediatric populations, this has translated to increased lengths of ICU stays, lengths of hospital admissions, numbers of ventilator days, and numbers of surgeries [3]. Dysfunction can be manifested by decreased left ventricular ejection fraction, stroke volume, cardiac output, and increased metabolic demand. At the cellular level, processes such as reduced cardiomyocyte contractility and disruption in calcium handling may drive this process. Right heart failure was found to be the most common cause of death in severely burn-injured infants in a 15 year retrospective study at a major pediatric burn center [4]. Rare cases of dilated cardiomyopathy following burn injury have been reported [5].

Burn-induced cardiac dysfunction occurs as early as 8 hours after injury and may last for up to 2 years [6]. The inflammatory cascade has been shown to alter immune related gene expression in blunt trauma and endotoxemia [7,8,9,10]. Alterations in gene expression may also drive burn-induced cardiac dysfunction and lead to persistent cardiac stress. Transcriptomics studies are needed to characterize the changes in gene expression at the tissue specific level following burn and traumatic injury. These studies may provide information on therapeutic targets and potential biomarkers for the identification of patients at risk for cardiac dysfunction. The purpose of this review is to summarize the mechanisms of a selection of cytokines in burn-induced cardiac injuries. This review was not meant to be an exhaustive compilation and specific cytokines were selected given the larger bodies of knowledge associated. We additionally aim to summarize the knowledge on changes in cytokine genome expression in response to inflammation and, more specifically, burns.

## 2. The Role of Cytokines in Burn-Induced Cardiac Dysfunction

Cytokines are low-molecular weight proteins secreted by cells responding to a variety of environmental stimuli. These proteins bind to membrane-bound or soluble membrane receptors leading to downstream changes in biological functions. There are a multitude of cytokine sources in the body including macrophages, T-lymphocytes, neutrophils, fibroblasts, natural killer cells, endothelial cells, and others.

Cytokines have been implicated in the pathogenesis of cardiovascular dysfunction in endotoxemia [11], trauma [12], burns [13], ischemic heart disease [14], cardiomyopathies [13], and heart failure [15]. Burn injuries specifically produce a persistent inflammatory state promulgated by the release of inflammatory mediators from the initial injury site. Cytokines play a role in this inflammatory cascade, exerting their effects on distant sites from the initial injury and promoting the further release of inflammatory mediators [2]. The mechanisms by which cytokines cause dysfunction include the dysregulation of calcium homeostasis [13], alterations in nitric oxide synthase activity [16], and the inducement of oxidative stress [17]. The following discussion examines how these mechanisms are affected by specific cytokines. The cytokines reviewed included TNF-α, IL-1β, IL-6, and IL-10. A summary of the mechanisms discussed and the study model are provided in Table 1.

### 2.1. Tumor Necrosis Factor α (TNFα)

TNFα is primarily a pro-inflammatory cytokine that has been well studied in both burn [33] and non-burn [34] related cardiac dysfunction. This cell signaling protein exists in two forms [35,36]. TNFα begins as a transmembrane protein that interacts primarily at TNFR2 receptors. Alternatively, TNFα may undergo processing by the TNFα-converting enzyme (TACE). This converts the enzyme into a soluble form that may act on type 1 or, less-commonly, type 2 receptors [35,36]. Type 1 receptors (TNFR1) are found on all tissues in the human body and are the “key” receptor for the transmission of TNFα’s effects. Downstream, TNFR1 binds to the TNFR1-associated death domain adapter protein (TRADD) and can trigger the formation of three possible complexes—each activating pro-inflammatory states or programmed cell death [35] (Figure 1). Type 2 receptors (TNFR2) are found on immune cells and promote the anti-inflammatory bioactivity of TNFα. TNFR2 receptors promote cell proliferation and survival as well as tissue regeneration [35]. Following activation by membrane-bound TNFα, TNFR2 recruits TRAF1, TRAF2, cIAP1, and cIAP2. This complex then activates protein kinases such as MAPK and protein kinase B in addition to transcription factors such as NF-κB (Figure 2).

TNFα in non-burn cardiac models has been shown to reduce left ventricular function [36,37]. Studies on cultured cardiomyocyte lines have shown a reversible dose dependent decrease in contractility [36,38]. This relationship is also seen in vivo [37,39,40]. In burn cardiac models, increased TNFα expression has been identified within 1 h of burn injury and left ventricular dysfunction has been found to occur at 8 h post-injury, persisting beyond 24 h [34,41,42,43]. Giroir identified higher expression of TNFα by cardiomyocytes in comparison to other tissue lines [44]. These studies demonstrate that TNFα alters cellular function. They additionally allude to the inflammatory cascade’s ability to upregulate cytokine secretion at tissue specific locations.

One cellular process by which TNFα may cause cardiac dysfunction is through the disruption of calcium homeostasis [18,19,45]. Yokoyama found decreases in cardiomyocyte intracellular calcium concentrations following administration of TNFα. This process may be caused by synergistic action with IL-1 beta leading to altered ryanodine receptor 2 (RyR2) function causing increased an frequency of spontaneous calcium release and asynchronous triggered calcium release [18,20,45]. Calcium homeostasis may be further altered by the dysregulation of the sarcoendoplasmic reticulum calcium ATPase (SERCA-2) pump [27]. The SERCA-2 pump is regulated by the pump inhibitory protein phospholamban. Phosphorylation of phospholamban by protein kinase A at Ser16 leads to disassociation from the SERCA-2 pump and allows calcium uptake into the sarcoplasmic reticulum. Protein phosphatase-1 (PP1) dephosphorylates phospholamban, which promotes the binding and inhibition of the SERCA-2 pump. The protein kinase C isozyme PKCα promotes upregulation of PP1 [46]. Treatment of rat cardiomyocytes with TNFα have shown increased membrane activation of PKCα and PKCε [21,22].

Another theorized mechanism for TNFα induced cardiac dysfunction is through increased inducible nitric oxide synthase activity. Nitric oxide synthase exists in three isoforms, namely neuronal NOS (nNOS), endothelial NOS (eNOS), and inducible NOS (iNOS). nNOS and eNOS are constitutively active isoforms. As the name suggests, the transcription of iNOS is induced by pro-inflammatory cytokines. Nitric oxide (NO) exerts two effects: the activation of guanylate cyclase to produce cyclic GMP (cGMP) and the post-translational nitrosylation of proteins [23]. cGMP participates in the signaling of negative inotropic processes within the cardiomyocyte through protein kinase G (PKG). There is limited knowledge regarding the role of deranged cGMP signaling in cardiac dysfunction. The post-translational nitrosylation of proteins contributes to the accumulation of reactive oxygen (ROS) and reactive nitrogen species resulting in intracellular injury. Finally, nitric oxide has been found to increase the expression of pro-apoptotic proteins of the Bcl2 family—Bax and Bak.

The utility of TNFα antagonists in cardiac dysfunction remains questionable. Studies in burn models are limited. In a 1994 study by Giroir, a human TNFα antibody was found to improve left ventricular function in burn injured guinea pigs [47]. Later, results in a phase I study by Deswal on NYH class III heart failure patients using etanercept—a soluble TNFα receptor antagonist—showed borderline significant improvements in ejection fraction after a 6 minute walk test and reduced serum levels of TNFα [24]. However, the study was limited in size. Subsequent large-scale trials on NYH class II and III heart failure patients by Mann (RECOVER and RENAISSANCE) showed etanercept produced no clinically significant improvements in outcomes [48]. In the ATTACH trial, infliximab—a chimeric monoclonal antibody composed of a murine Fab fragment combined with a human IgG Fc fragment—was examined. A dose-dependent increase in hospitalizations for heart failure and all-cause mortality were identified [28]. The lack of clinical benefit may be explained by the dual pro-inflammatory and anti-inflammatory role of TNFα. TNFα antagonists indiscriminately block both processes as they compete for the ligand. Thus, the anti-inflammatory processes needed for healing are inhibited. Furthermore, it is difficult to determine the generalizability of heart failure studies to burn models. The circulating levels of TNFα may be higher in burn models given the severity and acuity of injury. Further knowledge is needed in identifying the clinical utility of TNFα targeted therapies in burn-induced cardiac dysfunction.

### 2.2. Interleukin 1 β (IL-1β)

IL-1β is a potent pro-inflammatory and pyrogenic cytokine of the IL-1 family. Although two receptors for IL-1β have been identified, only one affects cellular processes. The first receptor type—IL-1R1—may activate the transcription factor NFκβ or mitogen-activated protein kinases (MAPK) including JNK, p38 MAPK, and p42/44 MAPK. The second receptor type—IL-1R2—is a decoy receptor that does not transduce downstream signals [49] (Figure 3).

IL-1β has been shown to reversibly reduce left ventricular function in non-burn animal models by reducing β1 adrenergic responsiveness [50,51]. Studies by Van Tassell showed that this reduction in left ventricular function reversed when IL-1β was eliminated or increasing doses of isoproterenol were administered [51]. These data suggest that the cardiac dysfunction caused by IL-1β is more likely related to functional impairment in β1-receptor stimulation rather than structural change [52]. Furthermore, the mechanism of reduced ventricular function may be due to decreased intracellular cyclic AMP from reduced β1-receptor responsiveness [50]. A study on rat cardiomyocytes treated with IL-1β for 3 days demonstrated a decrease in cAMP production and contractile response to adrenergic stimulation compared to controls. These results were reversible when cells were grown in cytokine free media for 3 days. Conflicting evidence does exist. A study by Liu et al. found no decrease in intracellular cAMP one hour after the administration of IL-1β in rat ventricular cardiomyocytes [53]. L-type calcium channels are down-stream activated through cAMP phosphorylation. In the presence of IL-1β, L-type calcium channel phosphorylation was not affected. These results may have been a product of inadequate incubation time with IL-1β as previous studies looked at longer incubation points.

As with other cytokines at the level of the heart, IL-1β may cause disruption in calcium homeostasis. IL-1β and TNFα have been found to work synergistically to reduce calcium concentration in the sarcoplasmic reticulum by increasing the calcium leak and asynchronous release [19,20,45]. Additionally, in a study by Combes, exposure of cardiomyocytes to IL-1β for 3 days led to a decreased gene expression of SERCA, phospholamban, and L-type calcium channels [25].

Finally, similar to TNFα, IL-1β has been found to increase inducible nitric oxide synthase activity [18,20,45]. This leads to increased intracellular cyclic GMP, reducing cardiac contractility through a PEDE5A-CGMP-PKG pathway previously studied in our lab [54].

In burn injured models, most IL-1β release occurs at the site of the burn injury [43,55]. Similar to TNFα and IL-6, IL-1 beta expression rises within 1 hour of burn injury and may remain elevated for over 48 hours [42]. Cardiac dysfunction has been documented to occur as early as 8 hours following burn injury [42].

IL-1β and its downstream pathways have been tested as potential targets in non-burn cardiac dysfunction secondary to ischemia/reperfusion (I/R) injury and cardiomyopathy. Rosuvastatin therapy in combination with ticagrelor—a platelet aggregation inhibitor—has been found to reduce cardiomyocyte apoptosis in I/R injury and infarct size. A concomitant decrease in IL-1β mRNA was found in combination with these findings (Birnbaum et al., 2016). In murine models, IL-1 antagonists were shown to decrease infarct size following I/R injury and preserve ejection fraction after cardiac injury [26,56]. The applicability of these studies to burn models is uncertain and limited knowledge on therapies targeting IL-1β in burn populations exists. A recent study in our lab found that phosphodiesterase inhibitors Sildenafil [52,57], MITO-TEMPO [41], and Oltipraz [58] normalized burn-injury associated increases in IL-1β, thus suggesting that the PDE5A-cGMP-PKG pathway, cardiac mitochondrial phosphorylation pathway, and Nrf2-ARE pathway are involved in burn injury-induced IL-1β secretion.

### 2.3. Interleukin-6 (IL-6)

IL-6 is a pleotropic and pro-inflammatory cytokine that is secreted by a wide variety of cells throughout the body. IL-6 exerts its effects through the IL-6 receptor (IL-6R) and gp 130. Due to its long half-life, IL-6 remains in the serum long after other pro-inflammatory cytokines have dissipated [59,60]. This has made it a useful marker for critical illnesses. Elevated serum levels of IL-6—in addition to IL-8—have been associated with increased mortality in patients with severe burns [61,62].

Limited knowledge exists regarding the exact role of IL-6 in cardiac dysfunction. IL-6 levels have been found to be elevated in mice with cardiac R/I and cardiomyopathy [63,64,65,66]. However, in burn models, IL-6 was demonstrated to be less potent in reducing cardiac contractility than other inflammatory mediators, specifically TNFα or IL-1 beta. Combined, all three cytokines synergistically have been found to decrease left ventricular function [42]. Taken together, this evidence raises the question as to whether IL-6 merely amplifies inflammatory activity or directly acts on pathways affecting cardiac contractility.

There is evidence to suggest that IL-6 plays a role in calcium handling following burn injury [67,68]. Calcium and sodium concentrations in cardiomyocytes were compared in mice overexpressing IL-6 and IL-6 knockout mice. Following burn injury, mice overexpressing IL-6 demonstrated significantly higher calcium and sodium levels compared to knockout and wild type mice, respectively. Decreased expression of the sarcoplasmic reticulum calcium ATPase (SERCA) in response to IL-6 may explain this finding. SERCA is an important exchange protein involved in the recycling of calcium from the cytosol back to the sarcoplasmic reticulum following cardiomyocyte excitation. Reduced SERCA activity leads to increased cytosolic calcium levels. This increases the activity of the sodium–calcium exchangers at the cell membrane. Villegas et al. found that IL-6 exposure decreased both SERCA mRNA and protein in cultured rat cardiomyocytes [69].

IL-6 may alternatively cause decreased cardiac contractility through the enhancement of iNOS activity. The exposure of isolated hamster papillary muscles to IL-6 reduced contractility. This effect was reversed with the addition of a nitric oxide synthase inhibitor. Further work has demonstrated that the enhancement of iNOS activity is likely mediated through the STAT3 pathway [70]. Continued research is needed to understand the exact mechanism of IL-6 in cardiac dysfunction, which may provide future therapeutic targets.

Current therapeutic options targeting IL-6 and its related pathways have shown promise in treating cardiovascular dysfunction in R/I in non-ST segment and ST segment elevation myocardial infarction. In a phase 2 trial, tocilizumab—a monoclonal antibody of the IL-6 receptor—has been shown to increase myocardial salvage and attenuated the inflammatory response after injury [29,71]. The exact mechanism by which tocilizumab produces this effect is not fully understood and remains an area for further research.

### 2.4. Interleukin-10 (IL-10)

IL-10 is an anti-inflammatory cytokine released by a variety of cells. It mainly exerts its anti-inflammatory effects by reducing the inflammatory responses of macrophages [30], namely inhibiting antigen presentation by dendritic cells [72], macrophage activation, and macrophage infiltration into the injury site. IL-10 additionally inhibits apoptotic signaling in cells [73].

Limited studies have been performed on IL-10′s impact on cardiac function in burn injured patients [74]. In non-burn cardiac models, IL-10 has been found to be cardioprotective [75]. In vivo studies of hypertrophic cardiomyopathy in mice showed reduced fibrosis, apoptosis, pro-inflammatory gene expression, and inflammatory signaling when treated with IL-10 [76,77]. While these anti-inflammatory properties can attenuate the body’s response following trauma, they can prove detrimental if the immune system becomes too suppressed to resist infection. Studies have found that IL-10 levels are significantly higher in burn patients who died from subsequent burn sepsis. IL-10 levels can thus be considered as a predictive model for sepsis mortality in patients with burn injury [78,79].

## 3. Changes in the Cytokine Transcriptome in Inflammation

Following burn injury, inflammatory mediators are released from the injury site into the systemic circulation. These mediators may then promote dysfunction and changes in gene expression at distant tissue sites. An early study on the transcriptome response to inflammation was conducted by Calvano et al. in 2005 [7]. Bacterial endotoxin was given to volunteers and whole blood was sampled at 2, 4, 6, 9, and 24 h timepoints. Leukocyte RNA analysis and gene ontology revealed changes in the expression of transcription factors as early as 4–6 h post injection. Notably, protein synthesis expression and mitochondrial energy production were significantly downregulated [7]. Subsequent studies have shown changes in genome expression occur following traumatic injury and burns [2,8,9,10,80]. Furthermore, cytokine synthesis has been found to increase in response to environmental stress. p38/MAPK—a pathway involved in the regulation of cytokine synthesis—was found to be upregulated in whole blood leukocytes following blunt trauma [8]. While these studies demonstrate transcriptome changes in response to inflammation, the focus was placed on whole blood leukocytes creating non-specific results for distant tissue sites.

Changes in cytokine gene expression in response to inflammation have been studied in a limited number of tissue sites [31,32,81,82]. In skeletal muscle, the expression of TNF-alpha, INF-gamma, IL-17A, IL-6, CCL5, and TGF-beta were increased in patients with peripheral arterial disease (PAD) [31]. PAD is severe atherosclerosis of the lower extremities, causing chronic intermittent ischemia. Inflammation ensues at the affected muscle levels. Differential gene expression in keratinocytes has been examined following burn injury. Studies have found upregulation of the cytokines IL1A, IL1B, IL5, IL8, and IL37. The chemokines CCL2, CCL7, CCL11, CCL 17, CCL20, CCL 25, CCL 26, CCL27, CXCL 1, CXCL 3, CXCL 5, CXCL 6, CXCL 12, and CXCL 14 were also found to be upregulated. Chemokine receptors CCR1 through CCR 5 in addition to cytokine receptors IL13, IL36G, and LTB [32,81,82]. In asthmatics, IL-1, Il-5, IL-6, IL-25, TNF-alpha, and NF kappa beta expression were upregulated in sputum samples [83,84]. This indicated differential gene expression was occurring in lung epithelial cells. The heterogenous expression of cytokines across the tissue sites mentioned above indicate the differential expression of cytokines occurs in response to inflammation. There is some overlap in upregulated gene expression. However, varied findings also indicate tissue specific changes in cytokine regulation. This limits the generalizability of findings. Understanding the nuances of cytokine expression based on the tissue site may aid in understanding the complexities of the inflammation cascade particularly in burn models.

In the heart, few transcriptome studies have been performed in burn models. Ballard-Croft et al. in 2001 looked at p38/MAPK expression in murine cardiomyocytes following burn injury [2]. This re-demonstrated the results seen in whole blood leukocytes of blunt trauma patients. Our lab previously found 38 up-regulated and 19 down-regulated genes related to Toll-like receptor (TLR) signaling in rat cardiomyocytes following burn injury [70]. Burn injury is an initially sterile process, thus no microbial components, such as LPS, circulate and activate pattern recognition receptors to promote inflammation. HMGB1 is a chromatin-associated protein that is released from necrotic cells into the cytosol and extracellular space contributing to inflammation in sterile conditions [85,86]. HMGB1 may then act on TLR4 to induce cytokine production [85,86]. Our study identified higher upregulation of TLR2, TLR4, and HMGB1 in cardiomyocytes [41]. As of this review, no other studies have examined the changes in cytokine expression in cardiomyocytes post burn injury. Further study is needed that may lead to the identification of potential therapeutic targets or biomarkers and provide information on patients at high risk for cardiac dysfunction following burn injury.

It is not well established how patient specific factors such as TBSA, gender, and age affect tissue-specific cytokine expression or how long these alterations last. A small study in 2014 on human keratinocytes showed greater upregulation of IL-8 and IL-1R1 in large burns (25–50% TBSA or 10% requiring skin grafting) compared to small burns (<5% TBSA) (Gragnani et al., 2014). IL-8, which was not discussed in our previous section, is a pro-inflammatory cytokine involved in immune cell recruitment. These results reiterate the increased inflammatory response seen in larger burns (Jeschke et al., 2007).

Gender has been found to influence inflammation. Female gender has been found to be associated with favorable outcomes in trauma, hemorrhagic shock, and sepsis [87,88]. This is thought to be due to a combination of hormonal and epigenetic factors. Estrogen has a pro-inflammatory effect while testosterone is generally anti-inflammatory [87]. Queen et al. studied the effect of gender on the expression of cytokines and other inflammatory genes in mice exposed to LPS [89]. A limited number of genes were examined—IL-6, MIP-1β, IL-1β, Fgb, and Mt-1. A dimorphic pattern in expression was seen with IL-6 and Fgb in three inbred strains of mice. The female mice demonstrated higher expressions [89]. Additional studies are needed to identify the effects of gender on cytokine expression as they can elucidate the nuances of the inflammatory cascade.

There is evidence to suggest age influences the inflammatory response. Immune dysregulation has been found to occur with advancing age characterized by decreased antigenic reactivity and the increased production of pro-inflammatory mediators, indicators of immune dysregulation. Age-related increases in pro-inflammatory cytokines such as TNFα, IL-1 beta, and IL-6 have been found in healthy individuals [90]. However, in states of stress such as burns, the immune response in the elderly has been characterized by a delayed hyperinflammatory response and slowed healing [91]. This places them at a higher risk for infection, multisystem organ failure, and mortality. Changes at the level of gene expression have been identified. A study on elderly patients found that pro-inflammatory gene expression was attenuated early following traumatic injury. Decreased levels of IL-6, IL-8, IL-10, MCP-1, and TNFα were identified in the serum. Inflammatory gene expression increased by 4 days post injury; however, this was around the same timepoint that inflammatory gene expression began to normalize in younger patients [92]. In burn models, a similar dysregulated inflammatory response has been seen. Elderly patients demonstrated a predominantly downregulated immune response following burn injury in a transcriptome study by Dreckmann [93]. Upregulated genes were involved in destructive processes including protein degradation, complement activation, and hemolysis [93]. The elderly experience prolonged recovery periods and a higher risk of complication following burn injury compared to the young. Thus, further exploration of the influence of age on the cytokine transcriptome response in burn injury is needed.

## 4. Conclusions

In summary, burn injury causes a hyperinflammatory response that comprises multiple mediators including cytokines. Specific cytokines such as TNFα, IL-1 beta, IL-6, and IL-10 have been found to be involved in cardiac dysfunction following burn injury. The mechanisms by which these cytokines cause cardiac dysfunction include the dysregulation of calcium and sodium homeostasis and increased nitric oxide synthase activity. Therapeutic agents targeting these cytokines were explored.

There is evidence that inflammation causes transcriptomic changes systemically and, to a limited degree, tissue-specific sites. Burn injury induces a persistent, hyper-inflammatory response that has been found to have downstream effects on cytokine genome expression systemically and at tissue specific levels such as cardiomyocytes. Additional factors that may affect genome expression in inflammation include burn/injury size, gender, and age. However, more knowledge is needed in these areas to understand the full effect on cytokine expression and the inflammatory cascade. Continued study of the genome response in burn and in burn-induced cardiac dysfunction may provide risk factors that may be used by clinicians to identify patients at high risk for complication. Additional therapeutic agents and biomarkers may additionally be identified. 

## Figures and Tables

**Figure 1 jpm-12-01876-f001:**
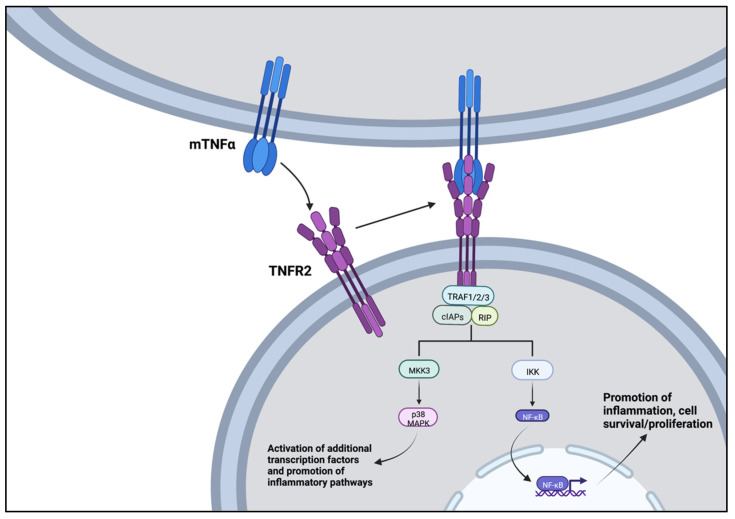
Activation of TNFR2 via membrane-bound TNFα (mTNFα). After activation of the TNFR2 receptor, a signal transduction complex formed by TRAF (1/2/3), cIAP(1/2), and RIP. The downstream pathways promote cell proliferation, cell survival, and tissue regeneration. Image created using BioRender.

**Figure 2 jpm-12-01876-f002:**
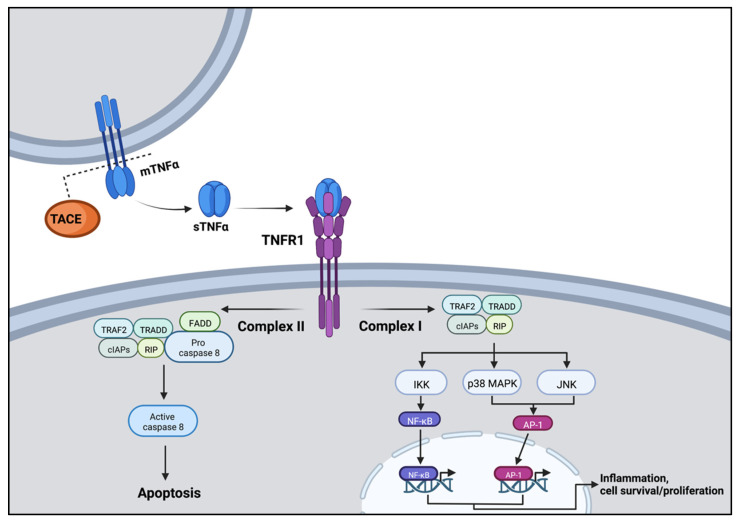
Activation of the TNFR1 receptor by soluble TNFα (sTNFα). Activation of TNFR1 results in recruitment of TNFR1-associated death domain adapter protein (TRADD). Different complexes may then arise based on protein interactions resulting in downstream activation of pro-inflammation and programmed cell death. Image created using BioRender.

**Figure 3 jpm-12-01876-f003:**
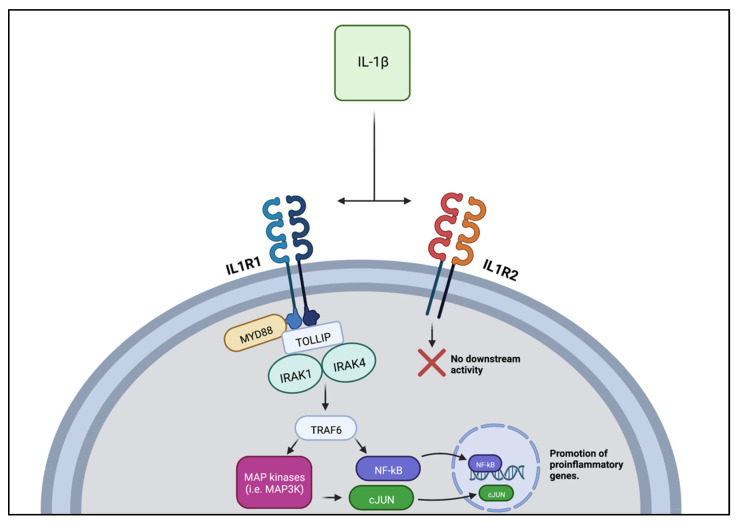
IL-1β activity at the IL1R1 vs. IL1R2 receptors. Activation of the IL1R1 receptor results in pro-inflammatory and pyrogenic activity via activation of transcription factors and protein kinases. The IL1R2 receptor is a decoy receptor with no downstream biological activity.

**Table 1 jpm-12-01876-t001:** Summary of discussed cytokines and mechanisms involved in burn-related cardiac dysfunction.

Cytokine	Mechanisms in Cardiac Dysfunction	Studied Model	Authors
TNFα	Increased Ryanodine receptor (RyR2) calcium release from sarcoplasmic reticulum. Increased cell membrane activation of PKCα: upregulation of protein phosphatase-1 (PP1) leading to inhibition of phospholamban and SERCA-2 activity. Increased Ca^2+^ independent nitric oxide synthase activity: increased cyclic GMP.	Rat, feline cardiomyocytes Rat, mouse cardiomyocytes Rat cardiomyocytes	Duncan et al., 2007 [18] Duncan et al., 2010 [19] Yokoyama et al., 1993 [20] Braz et al., 2004 [21] Jude et al., 2018 [22] Tan et al., 2007 [23] Schulz et al., 1992 [24]
IL-1β	Reduced β1 adrenergic responsiveness leading to reduced intracellular cyclic AMP and L-type calcium channel phosphorylation. Increased Ca^2+^ leak and asynchronous leak at the sarcoplasmic reticulum. Increased Ca^2+^ independent nitric oxide synthase activity: increased cyclic GMP. Decreased SERCA mRNA and expression of phospholamban, L-type calcium channels.PEDE5A-CGMP-PKG pathway.	Rat cardiomyocytes Rat, feline cardiomyocytes Rat, feline cardiomyocytes Rat cardiomyocytes Rat cardiomyocytes	Gulick et al., 1989 [25] Combes et al., 2002 [26] Duncan et al., 2007 [18] Duncan et al., 2010 [19] Yokoyama et al., 1993 [20] Duncan et al., 2007 [18] Duncan et al., 2010 [19] Yokoyama et al., 1993 [20] Zuo et al., 2018 [27] Combes et al., 2002 [26] Wen et al., 2020 [28]
IL-6	Decreased SERCA expression. Enhanced inducible nitric oxide synthase activity through the STAT3 pathway.	Mouse cardiomyocytes Hamster papillary muscle	Villegas et al., 2000 [29] Yu et al., 2003 [30]
IL-10	Anti-inflammatory, suppresses up-stream and downstream signaling of inflammatory cytokines.	Human serum	Lyons et al., 1999 [31] Pileri et al., 2008 [32]

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
