# Peer review of "Cytokine Pathways in Cardiac Dysfunction following Burn Injury and Changes in Genome Expression"

_jpm, 2022, doi:10.3390/jpm12111876_

Round 1

Reviewer 1 Report

Reviewer´s Comment

It is indeed an interesting piece of article stating the role of cytokine induced inflammation and detrimental outcome on cardiac dysfunction during burn injuries. The authors have well-structured the article into readable subtopics.

However, the story line brings a generalized effect while reading the manuscript and not stating any hypothesized or reported mechanisms. Instead, authors must focus on describing finely-tuned information in several subtopics (as mere references are not enough)

1. In the paragraph showing the role of IL-6, the homeostasis behind calcium and sodium levels upon IL-6 cytokine must be explained in detail. Moreover, the wound healing role of IL-6 should also be explained and must state the reason on why IL-6 cannot perpetuate would healing despite increased levels among these patients.

2. The schematic sketch looks very superficial and requires more details on the IL-1ß, TNF-a related pathway linking to cardiac dysfunction.

3. Under line no 274, the changes in cytokine gene expression during inflammation was generally stated and authors must improve the statement by describing the names of changed cytokines, whether upregulated or downregulated and type of inflammation to be mentioned here and furthermore brief description on the mechanism should also be written.

4. Next Authors should introduce a comprehensive table showing the role of TNFa, IL-1beta, IL-6 and IL-10 in inducing cardiac dysfunction where human and animal studies must be described separately, with appropriate references.

Author Response

Hello,

Thank you very much for your time and comments.

  1. The paragraph describing the role of IL-6 has been revised. Detail regarding calcium and sodium homeostatic alteration has been added.
    We are unsure of the second portion of this critique point. This review is focused on cytokines in cardiac dysfunction following burn injury. Discussion on IL-6 in relation to burn wound healing seems to be out of scope for this paper.
  2. Additional detail had been added to Figures 1 and 2 to further describe the associated pathways. Unfortunately, at this time, there is no specifically described downstream pathway linking IL-1ß or TNF-a’s activity to decreasing cardiac function. The figures were made as a general reference for the reader.
  3. The section regarding changes in cytokine expression in relation to inflammation has been adjusted. Specific up and down regulated cytokines in different tissue sites are mentioned.
  4. A summary table has of the cytokine roles, study models, and associated studies has been included.

Reviewer 2 Report

1. Instead of simply saying "burn-induced cardiac dysfunction", it is better to have more details of cardiac changes following burn injury, i.e, symptoms of cardiac dysfunctions, such as changes in stroke volume, cardiac output, oxygen consumption, metabolic activity...

2.  Based on the information in the manuscript, authors intend to show TNFR1 pathway in Figure1 and TNFR2 pathway in Figure2. However, it turns out that Figure1 in the manuscript shows schematic of TNFR2 pathway, Figure2 shows schematic of TNFR1 pathway.

3.TRAF1, TRAF2, cIAP1 and cIAP2 are recruited to TNFR2 following receptor activation, so the direction of arrow symbol in the Figure1 should be toward to TNFR2 receptor.

4.One of cIAP1 symbols in Figure1 should be cIAP2.

5.To better display signal transduction from membrane receptor to nucleus, it is better to draw symbol of nucleus in the cell in Figure1 and Figure2.

6.There are three Nitric oxide synthases, including constitutively expressed neuronal NOS (nNOS) and endothelial NOS (eNOS), and inducible NOS(iNOS). These three isoforms exhibit distinct functions and are encoded by different genes. To summarize influence of TNFa on Nitric oxide synthase, the authors need to clearly indicate that TNF α and IL-1 beta affect cytokine-inducible nitric oxide synthase(iNOS) and at least briefly summarize how iNOS leads to cardiac dysfunction(there are plenty of literature available). iNOS is a more accurate terminology than “calcium independent nitric oxide synthase”.

Author Response

Thank you for your time and review comments.

1. The introduction has been modified to include specific manifestations of cardiac dysfunction. 

2-4. Figures 1 and 2 have been edited to demonstrate the associated pathways. Additional detail has been added. 

5. A nucleus has been added to Figures 1 and 2 have to better demonstrate signal transduction.

6. Thank you for the clarification regarding isozymes of nitric oxide synthase. We have corrected the TNF alpha section and subsequent wording used throughout the manuscript.

Round 2

Reviewer 1 Report

Since Authors have fulfilled the required revision, I accept for the publication of manuscript in its current form.